# Performance of Composite Portland Cements with Calcined Illite Clay and Limestone Filler Produced by Industrial Intergrinding

Edgardo F. Irassar * , Viviana L. Bonavetti, Gisela P. Cordoba , Viviana F. Rahhal, Claudia Cristina Castellano and Horacio A. Donza

Facultad de Ingeniería, CIFICEN (UNCPBA-CICPBA-CONICET), Universidad Nacional del Centro de la Provincia de Buenos Aires, Olavarría B7400JWI, Argentina
* Correspondence: firassar@fio.unicen.edu.ar; Tel.: +54-2284-451-055 (ext. 230)

**Abstract:** The performance of five composite Portland cements (CPCs) with limestone filler (LF = 10%–25% by mass) and calcined illite clay (CIC = 10%–25% by mass) elaborated by intergrinding was analyzed in paste, mortar, and concrete. Hydration was studied by isothermal calorimetry, bound water, and XRD. Flow and compressive strength (2 to 90 days) were determined in standard mortar. Concretes (w/b = 0.45; binder content = 350 kg/m³; slump = 15 ± 3 cm) were elaborated to determine compressive and flexural strength, water penetration, and chloride migration. Intergrinding CPCs have a large specific surface area when LF + CIC increases, with a similar size range of clinker particles. Supplementary cementing material replacements decreased the heat rate, prolonged the dormant period, and decreased the acceleration rate at early ages. According to the Fratini test, all CPCs had positive pozzolanicity after 28 days, but XRD analysis showed $Ca(OH)_2$ associated with monocarboaluminate phases. Mortar flow was slightly reduced when the proportion of CIC was increased. Mortar strength decreased when the sum of LF + CIC increased. CPC strength class was limited by compressive strength after 28 days. Concretes were workable, and the compressive strength after 28 days depended on the LF + CIC, and CIC contributed after 90 days. After 28 days, the water penetration depended mainly on the LF + CIC content. The chloride migration coefficient was also reduced when CPC contained more CIC and less LF.

**Keywords:** illite; calcined clay; limestone; composite cements; intergrinding; strength





## 1. Introduction

According to the IPCC [1], the progress in industrial processes is insufficient to reduce global emissions due to the increase in cement demand associated with rapid growth and urbanization in developing countries. After several efforts of the cement industry to reduce $CO_2$ emission and increase energy efficiency during Portland clinker production, one-ton Portland clinker currently emits ~810 kg $CO_2$ due to the decarbonation of limestone (510 kg $CO_2$) and the use of fossil fuels in the kiln. Clinker also requires high embodied energy (3.4–3.5 GJ/t) [2].

The Global Cement and Concrete Association roadmap [3] stipulates the paths to attaining net-zero emissions by 2050 for the cement industry. For the construction sector, one strategy is to reduce the clinker factor in general-use cement by using a suitable combination of supplementary cementing materials (SCMs). The most widely used SCMs are limestone, fly ash, granulated blast furnace slag, and natural or artificial pozzolans.

Limestone is the most used SCM in blended cement, contributing to the completion of the particle size distribution curve, filling the spaces between particles and improving the packing density of the cement [4–6], thereby stimulating the early hydration of Portland cement [7–10], stabilizing ettringite due to the formation of monocarboaluminate [11,12], and improving the early compressive strength for low percentage replacement (<12%) [7,13,14]. At later ages, the dilution effect preponderantly limits the LF content [15,16] due to the increased capillary porosity that affects strength and durability [17,18].

Fly ash obtained from coal-fired power plants and granulated blast furnace slag from steel production have been widely proven to improve late strength and durability [19]. However, the latter SCMs are scarce in Latin America [20], while regional availability limits the use of natural pozzolans [21,22]. Artificial pozzolans are a potential solution to all these drawbacks [23–25].

In the central region of Buenos Aires Province, the Tandilia system has a crystalline basement (gneiss, granitoid, and migmatite) and two superimposed packages of marine sedimentary cover (one Neoproterozoic and the other Lower Paleozoic). The Neoproterozoic profile presents several layers of sandstone, dolomite, claystone, and limestone [26]. Near Olavarria, the availability of limestone has supported the development of the largest cement production plants in Argentina. The limestone mantle occurs between two claystone layers. Depending on the depth and location of the deposit, clays are composed of different minerals (illite, smectite, chlorite, and (to a lesser extent) kaolinite) associated with quartz, iron oxide, feldspars, carbonates, and anatase [27]. Shales formed by illite are the most abundant in this region, while primary kaolinite clays are found in reduced layer thickness for use as SCMs [28].

According to this geological setting, illite shales have emerged as a suitable alternative to produce an artificial pozzolan by thermal activation [29,30]. Depending on the clay minerals, thermal activation of clay occurs at 550 to 950 °C—temperatures below that of clinkerization (1450 °C) [31]. This treatment implies lower embodied energy and lower $CO_2$ emission than those produced by clinker. For different clays, thermal treatment is performed to achieve the complete dehydroxylation of main clay minerals, producing an amorphous phase that can react with $Ca(OH)_2$. The potential of calcined clays as artificial pozzolan has been studied by multiple researchers [32–35].

Several composite Portland cements combining two SCMs have been studied to produce a general-use cement, such as slag limestone [14,36,37], natural pozzolan slag [38] fly ash limestone [39,40], and, recently, LC3 cement, which is based on calcined kaolinite clay and limestone [41–43].

In this paper, we describe the properties of composite Portland cement (CPC) made with a combination of limestone filler (10%–25%) and calcined illite clay (10%–35%) in paste, mortar, and concrete. The characteristics and hydration of CPC were analyzed, and the workability, mechanical strength, and 28-day transport parameters of concrete samples were compared.

## 2. Materials and Experimental Procedures

### 2.1. Materials

For composite Portland cements (CPCs), a Portland clinker (CK), natural gypsum (NG), limestone filler (LF), and calcined illite clay (CIC) were used. For these materials, the average chemical and mineralogical compositions obtained during industrial trial production are given in Table 1.

Typical clinker from the Pampa region of Argentina is produced using micritic limestone, low-aluminum, and high-iron clays as raw materials in a plant using a dry process with a precalciner using natural gas as fuel. Clinker has high $C_3S$ content (68.9%) and low $C_3A$ content (2.9%). Natural gypsum from Rio Negro Province contains 85.2% of gypsum, and impurities (quartz and illite). Limestone from the cement factory is mainly composed of calcite ($CaCO_3$ = 88.2%), microcrystalline quartz, and some traces of clays (illite and kaolinite).

Calcined clay was produced in an industrial pilot trial using claystone (also called shale) obtained from a quarry located in the SW of the Tandilia system (Olavarría, Buenos Aires, Argentina) as raw material. The thick layer of illite claystone (illite shale) of marine origin in the Cerro Largo Fm is roofed by the micritic limestone of the Loma Negra unit [26]. The geological description shows that claystone is massive limonite with finer grain, with color ranging from red to purple, and is a non-plastic material (Plastic Index < 2.6). It is classified as a ferruginous shale. The main mineral is illite with fine anhedral quartz and a

low proportion of clastic quartz impregned by iron oxide. Figure 1 shows the XRD pattern of raw illite claystone, and the average mineralogical composition determined during the industrial trial pilot using the Rietveld method was illite (60%), quartz (34%), feldspars (3%), and hematite (1.5%), with less than 1% chlorite, kaolinite, and calcite.

**Table 1.** Chemical and mineralogical composition of materials.

| Element/Mineral | Clinker | Natural Gypsum | Limestone | Calcined Illite Clay |
|---|---|---|---|---|
| Chemical composition, % | | | | |
| $SiO_2$ | 21.1 | 1.6 | 9.1 | 63.2 |
| $Al_2O_3$ | 4.4 | 0.4 | 1.0 | 19.6 |
| $Fe_2O_3$ | 5.1 | 0.2 | 0.6 | 8.1 |
| CaO | 64.7 | 37.8 | 49.0 | 0.8 |
| MgO | 0.7 | 0.5 | 0.5 | 1.5 |
| $SO_3$ | 0.4 | 40.8 | 0.2 | |
| $K_2O$ | 0.9 | 0.1 | 0.3 | 5.6 |
| $Na_2O$ | 0.0 | | 0.0 | 0.1 |
| $TiO_2$ | 0.2 | | | 0.8 |
| LOI | 0.1 | 18.3 | 37.6 | 0.4 |
| Mineralogical composition, % | | | | |
| $C_3S$ | 68.9 | | | |
| $C_2S$ | 11.4 | | | |
| $C_4AF$ | 15.5 | | | |
| $C_3A$-cubic | 1.0 | | | |
| $C_3A$-orto | 1.9 | | | |
| Arcanite | 0.9 | | | |
| Aphthitalite | 0.2 | | | |
| Calcite | | 7.5 | 88.2 | |
| Quartz | | 1.0 | 9.9 | 36.0 |
| Gypsum | | 83.6 | | |
| Basanite | | 3.8 | | |
| Anhydrite | | 1.0 | | |
| Dolomite | | 2.7 | 0.5 | |
| Illite | | | 1.1 | 3.0 |
| Hematite | | | | 5.0 |
| Amorphous | | | | 56.0 |

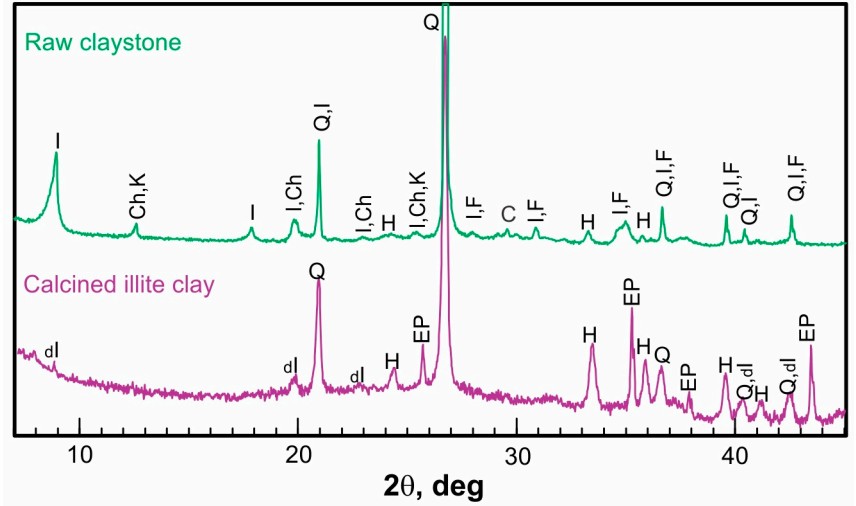

**Figure 1.** XRD patterns of raw claystone and calcined illite clay from an industrial pilot trial. I = illite; Ch = chlorite; K = kaolinite; Q = quartz, H = hematite; F = Feldspar; C = calcite; dI = dehydroxylated illite; EP = external patron.

After extraction and homogenization at the quarry, claystone was reduced in the jaw crusher and sieved into 10 to 50 mm size particles. Then, it was thermally treated in a rotary kiln at the lime line factory. For this pilot test, the flame temperature of the kiln was 900 ± 50 °C, using natural gas as fuel. The calcination temperature was determined in laboratory studies previously reported for different compositions of illite claystone [29,30]. The residence time was sufficient to cause the collapse of the illite crystalline structure and cooling at the kiln exit. Finally, the production was stacked in outdoor piles. During continuous production, samples of calcined clay were taken to control the process using XRD and to measure the intensity of the illite reflections at 2θ = 8.82° and 19.82°. Finally, the calcined illite clay was characterized by its chemical and mineralogical composition (Table 1). Pozzolanic properties were determined on the calcined clay ground in a laboratory ball mill with a RoS45 < 12% by mass.

The main elements of the CIC are $SiO_2$ (~63.2%) and $Al_2O_3$ (~19.5%). It has a high content of $Fe_2O_3$ (~8.1%), as reflected by its reddish color, as well as $K_2O$ (~5.6%) and MgO (~1.5%) from the decomposed illite and low content of $Na_2O$ and CaO. The XRD pattern of CIC (Figure 1) shows low-intensity peaks of dehydroxylated illite (I), and the associated minerals were quartz (Q) and the neoformation of hematite (H).

For a replacement of 25% by mass, CIC had a positive pozzolanic activity after 7 days according to the Frattini test (EN 197-5), and the pozzolanic activity index with cement at 20 °C (EN 450) was 0.88 ± 0.04, 0.92 ± 0.02, and 1.00 ± 0.02 after 7, 28, and 90 days, respectively. The activity index with Portland cement (IRAM 1654-1) was 0.85 ± 0.04. This procedure is similar to that described in ASTM C595-Annex 1. Prisms (40 × 40 × 160 mm) were cast using a mortar with a constant flow (110%–115%) and equivalent paste volume (~30% CIC by mass), then cured for one day at 21 °C and after being demolded for 27 days at 38 °C.

## 2.2. Composite Portland Cements (CPCs)

The proportions of composite Portland cement (CPC) were selected using a factorial experimental design; the variables were LF (10%–25%) and CIC (10%–25%). Table 2 reports the composition of each CPC (in % by mass). The minimum and maximum replacement levels were selected to explore the compositional domain of CPC of EN and Argentine standards (6%–35%). The central point corresponds to fifty–fifty replacements of each SCM (C17LF17CIC), and the extreme point has a replacement of 50% SCM. Previous research analyzed an LF-CIC system using laboratory-mixed composite Portland cements [44].

**Table 2.** Compositions of the studied CPCs.

| Composite Portland Cement | Proportions of Composite Cement, % by Mass | | | |
|---|---|---|---|---|
| | Clinker | Natural Gypsum | Calcined Clay (CIC) | Limestone (LF) |
| C10LF10CIC | 75.0 | 5.0 | 10.0 | 10.0 |
| C17LF17CIC | 60.0 | 5.0 | 17.5 | 17.5 |
| C25LF25CIC | 45.0 | 5.0 | 25.0 | 25.0 |
| C10LF25CIC | 60.0 | 5.0 | 25.0 | 10.0 |
| C25LF10CIC | 60.0 | 5.0 | 10.0 | 25.0 |

For comparison, an OPC (CEM-I EN 197) was used. It was produced with the same clinker, with a gypsum content of 5% and limestone filler as a minor component (<5%), ground with the same target as CPCs.

The optimum gypsum content was determined for the central point (C17LF17CIC). Cement with low $SO_3$ (1.00%) was ground in an industrial ball mill (RoS45 < 6%). Then, ground gypsum (100% passed through a 45 μm sieve) was added to reach 1.50, 2.10, 2.75, and 3.30% of $SO_3$. Optimum $SO_3$ was investigated using isothermal calorimetry of paste (w/b = 0.40) at 20 °C. According to ASTM C1679, the occurrence time and intensity of the sulfate depletion peak determined the optimum gypsum content. Gypsum is optimal

when this peak occurs hours later and has a lower intensity than the silicate peak. Based on the ASTM C563 criterion, compressive strength at 24 and 48 h was also determined using standard sand mortar EN 197 (1:1, w/b = 0.30) and variable gypsum content. Results show that the optimum $SO_3$ range is 2.1 to 2.8%, and the gypsum content was set at 5.0% by mass for all CPC.

Clinker, gypsum, limestone, and calcined clay were interground to fulfill the grinding target (RoS45 < 9%) in an industrial ball mill with two cameras integrated into a closed milling circuit equipped with a high-efficiency separator in the cement factory. CPCs were produced without grinding aids.

### 2.3. Characteristics of CPC

For the CPCs, the chemical composition was verified by XRF and LOI, and mineralogical composition was determined by XRD. In CPC, quartz content increased with the addition of CIC acting as a complementary filler. The following physical characteristics were determined: RoS45 using the wet method (ASTM C430), specific surface area by the Blaine method (ASTM C204), and density (ASTM C188). The particle size distribution (PSD) was determined by a laser granulometer (Malvern Mastersizer 2000E; Malvern Panalytical, Malvern, UK) with a dry-dispersed device (Sirocco 2000M; Malvern Panalytical, Malvern, UK).

Packing density (Ø) was measured by the wet packing method (WPM) developed by Wong and Kwan [45], defined as the maximum concentration of solids that binder can reach when mixed with water at different w/b ratios by volume. For high w/b ratios, solids are dispersed as a suspension with a low concentration. For low w/b ratios, the added water is insufficient to fill the voids, leaving air trapped between the binder particles with "loose" packing. In this w/b ratio range, there is a w/b ratio for which the concentration of solids is maximum, known as the packing density.

The WFT represents the average thickness of water films coating the solid particles [46,47] and can be calculated as the ratio of excess water $(u')_w$ needed to fill the voids in the particles' skeleton and the specific surface area (SSA) of the CPC:

$$\text{WFT} = (u')_w / \text{SSA} \tag{1}$$

For standard mortar, the packing factor of EN sand is 0.656, the minimum void ratio is 0.524, and the paste volume includes the water and the binder. Then, the excess paste ratio $(u'_p)$ was obtained as the paste volume minus the voids of the standard sand skeleton. Then, the paste film thickness (PFT), which represents the average thickness of paste coating the sand particles, can be calculated as the excess paste divided by the specific surface area of sand (SSAs) [48].

$$\text{PFT} = u'_p / \text{SSA}_s \tag{2}$$

The mortar flow was assessed using the flow table (ASTM C1437) in the RILEM mortar (sand: binder = 3; w/b = 0.50) containing standard siliceous sand. Mortar compressive strength was evaluated on $40 \times 40 \times 160 \text{ mm}^3$ prims. The reported value is the average of six tests.

A Frattini test (EN 196-5) was performed for composite cements after 7, 14, and 28 days. To this end, 20 g of binder and 100 mL of distilled boiled water were placed in an airtight container at 40 °C. The supernatant liquid was filtered at the test age, and the concentrations of $OH^-$ and CaO were determined. Finally, the result was compared with the solubility curve of $Ca(OH)_2$ at 40 °C, indicating that the cement is pozzolanic when the point is below this curve.

### 2.4. Hydration of CPC

CPC hydration was studied in cement pastes (w/b = 0.40) without chemical admixtures at 20 °C through the heat of hydration, bound water, and X-ray diffraction. The heat of hydration was determined in a conduction isothermal calorimeter for 48 h. The bound water (Wn) was determined after 2, 7, 28, and 90 days. Wn is defined as the mass

difference losses by calcination minus the loss of ignition (LOI) of the corresponding CPC to address the LF content and the minor loss of others components. For this purpose, samples were weighed and oven-dried to a constant weight at $105 \pm 5$ °C, calcined to $950 \pm 50$ °C, cooled in a desiccator, and reweighed. After 28 days, the mineralogical composition was determined by XRD on finely ground paste.

*2.5. Concrete Properties with CPC*

Concrete mixtures were elaborated with 350 kg/m$^3$ of binder and a water-to-binder ratio (w/b) of 0.45. Natural silica sand with a density of 2.64 g/cm$^3$ and a fineness modulus of 2.02 was used as fine aggregate, and crushed granite with a maximum size of 19 mm and a density of 2.72 g/cm$^3$ was used as coarse aggregate. The fine-to-coarse aggregate ratio was 43%. The slump of concrete mixtures was kept at $150 \pm 25$ mm using a dosage (3.2%–3.5% by binder mass) of a polycarboxylic ether-based superplasticizer (SP). For the investigation plan, two batches of 0.12 m$^3$ of each concrete were made in a drum mixer in the laboratory.

Fresh concrete was evaluated by the slump test (ASTM C143) and the flow table test (EN 12350-5). Cylindrical specimens ($100 \times 200$ mm) were cast and compacted with a rod. They were kept in the molds in a laboratory environment, capped with high-strength mortar, removed from the mold after 40 h, and immersed in lime-saturated water at $20 \pm 1$ °C until the test ages (2, 7, 28, and 90 days). Prismatic ($75 \times 100 \times 450$ mm) and cubic (150 mm) specimens were kept in molds, demolded, and cured in lime-saturated water until the test age.

The compressive strength (ASTM C39) was evaluated in five $100 \times 200$ mm specimens after 2, 7, 28, and 90 days, while flexural strength (ASTM C78) was determined on two prismatic specimens using three-point disposition.

After 28 days of curing, the capillary water absorption (ASTM C1585) was measured on three slices of cylinders (Ø = 100 mm and h = 50 mm). The initial rate of water absorption was determined as the slope of the best-fit line to the mass gain by unit area versus the square root of time (up to 6 h), and the water absorption capacity was determined according to IRAM 1871 as the increase in mass per unit area of the sample when the mass variation was less than 0.1% between two successive measures. The water penetration (IRAM 1554) under pressure was determined on the lateral surface of 150 mm cubes using incremental water pressure (24 h @ 1.0 MPa, 48 h @ 3.0 MPa, and 24 h @ 7.0 MPa). Then, the specimen was split, and the water penetration depth was measured, recording the average, as well as maximum and minimum values.

The chloride migration coefficient in a non-steady state was determined according to NT Build 492 on two slices with 100 mm diameter and 50 mm height. Slices were extracted from the half-height of the 200 mm height specimens cured for 28 days. After completing the test time, the slices were split and sprayed with AgNO$_3$ solution, and the depth of chloride ingress was measured. The chloride migration coefficient ($D_{nssm}$) was determined according to Equation (3):

$$D_{nssm} = \frac{0.0239\,(273 + T)\,L}{(U - 2)\,t} \left( x_d - 0.0238\sqrt{\frac{(273 + T)\,L\,x_d}{U - 2}} \right) \qquad (3)$$

where $T$ is the average of the initial and final temperatures in the KOH solution (°C), $L$ is the slice thickness (mm), $U$ is the applied voltage (V), t is the test duration (hour), and $x_d$ is the average penetration depth (mm).

## 3. Results

*3.1. Particle Characteristics of Interground CPC*

The physical characteristics of CPCs and OPC are reported in Table 3. The retained-on 45 μm sieve (RoS45) was between 5% to 9%, in agreement with the grinding target (RoS45 $\leq$ 9%). The specific surface area (SSA) of CPCs is higher than that of OPC and increased

35 to 60 m$^2$/kg when CIC content increased from 10% to 25%; 20 to 50 m$^2$/kg when LF content increased from 10% to 25% by mass. CPCs density decreases proportionally to LF + CIC content since both SCMs are less dense than clinker.

**Table 3.** Physical characteristics and granulometric parameters of studied cements.

| Cement | Density | RoS45 | SSA Blaine | PSD Parameters | | | Packing Density (Ø) | Voids Ratio (*u*) | WFT | PFT |
|---|---|---|---|---|---|---|---|---|---|---|
| | | | | d10 | d50 | d90 | | | | |
| | | % | m$^2$/kg | μm | μm | μm | | | μm | μm |
| OPC | 3.12 | 7.5 | 360 | 3.31 | 20.01 | 51.62 | 0.624 | 0.615 | 0.74 | 9.43 |
| C10LF10CIC | 2.96 | 4.3 | 411 | 2.39 | 15.32 | 42.32 | 0.623 | 0.605 | 0.72 | 10.15 |
| C17LF17CIC | 2.93 | 5.3 | 410 | 2.14 | 15.14 | 44.87 | 0.619 | 0.617 | 0.71 | 10.29 |
| C25LF25CIC | 2.88 | 5.7 | 502 | 1.77 | 13.17 | 46.03 | 0.612 | 0.632 | 0.64 | 10.84 |
| C10LF25CIC | 2.94 | 8.6 | 446 | 2.23 | 15.10 | 46.21 | 0.606 | 0.650 | 0.62 | 10.24 |
| C25LF10CIC | 2.96 | 9.0 | 435 | 2.21 | 14.48 | 45.93 | 0.593 | 0.685 | 0.61 | 10.15 |

Figure 2 shows the granulometric curves as a cumulative retained percentage and by particle size fraction, and Table 3 reports parameters d90, d50, and d10. The PSD of CPCs is finer than that of OPC. The cumulative retained particle size curves are quite similar for all composite cements, and consequently, parameters d90, d50, and d10 are closer for all cements (~45, 15, and 2 μm, respectively). The fractional volume curve of particles shows that C10LF10CIC has a greater volume between 20 and 30 μm. In comparison, that volume decreases when the total percentage of SCM increases and the greater volume of particles is close to 30 μm. For higher percentages of SCM, the curve shows a shoulder at 5 to 7 μm, indicating the concentration of the finer particles of CIC or LF particles [49,50].

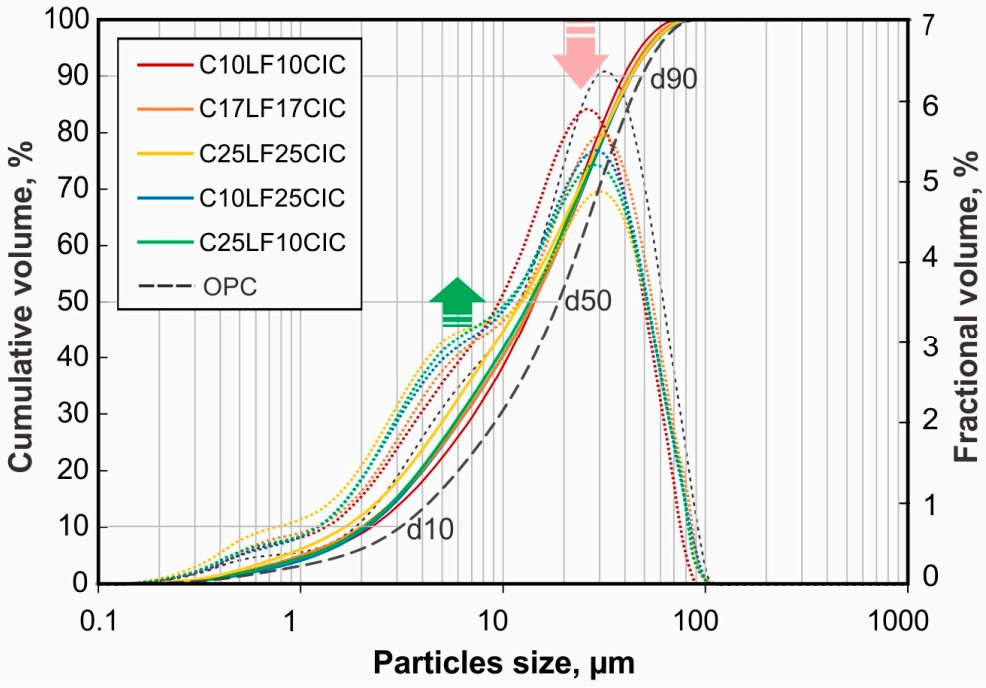

**Figure 2.** Particle size distribution of CPCs expressed in percentage of cumulative volume and percentage of fractional volume.

For the CPCs, the packing density (Ø) and the minimum voids ratio ($u'_{min}$) are reported in Table 3. C10LF10CIC and C17LF17CIC have a similar packing density to that corresponding to OPC cement (0.624), while C10LF25CIC and C25LF10CIC have a low packing density (0.606–0.593). For C25LF25CIC, the packing density is intermediate (0.612).

Using the packing density, the specific surface, and a w/b = 0.50 by mass, the calculated values of WFT for each CPC paste are reported in Table 3. The variation of WFT depends on Ø and the SSA of the binder [51]. C10LF10CIC and C17LF17CIC have a higher WFT than OPC because the excess water caused by the higher Ø is more significant than the proportional increase in the specific surface area. For C10LF25CIC and C25F10CIC, the WFT decreases due to the low Ø (large void fraction to be filled by water) and the high SSA, and both parameters could affect the flowability of the paste. However, the moderate changes in the WFT for the CPC are not significant enough to produce appreciable changes in the flow values.

The low density of CPC increases the paste volume in the standard mortar causing a thick film of paste surrounding the sand particles, which is calculated as PFT (Table 3). The low CPC density causes a thicker PFT than the OPC. Among the studied CPC compositions, the PFT has no significant changes.

### 3.2. Mortar Flow and Compressive Strength

The flow of standard mortar is shown in Figure 3. For CPC with 10% CIC, the increase in LF flow slightly increases (comparing C10LF10CIC with C25LF10CIC), while for CPC with 25% CIC (C10LF25CIC and C25LF25CIC), the flow decreases. For CPCs with 35% SCM, the flow decreases as the CIC content increases (C25LF10CIC > C17LF17CIC > C10LF25CIC). Concerning OPC, the flow of CPCs was lower than that obtained in paste using the mini-slump test [44,52].

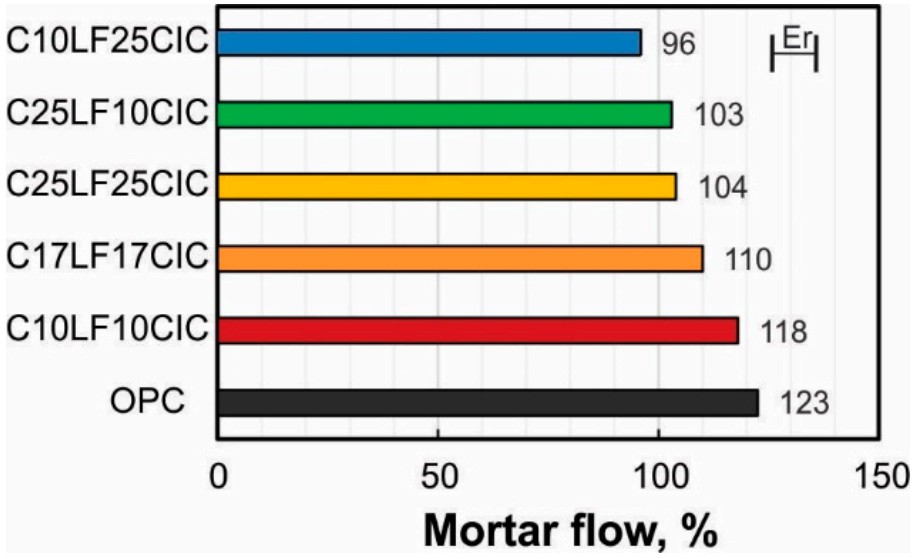

**Figure 3.** Flow of standard mortar for CPC and OPC.

Figure 4 shows the compressive strength (CS) of mortars with CPCs and OPC and the isoresponse curves at each test age. The dotted and dashed lines correspond to 80, 65, and 50% CS of OPC. After 2 days, the C10LF10CIC has the highest compressive strength (21.5 MPa) among CPCs, although it is lower than that of OPC (26.4 MPa). For CPCs with 35% SCM, C17LF17CIC has a higher strength (16.5 MPa) than that of C25LF10CIC (13.6 MPa) and C10LF25CIC (13.1 MPa), all of which are higher than the minimum required for the standard for a 42.5 strength class (10 MPa). The CS of C25LF25CIC (8.3 MPa) is less than 50% of the OPC and lower than the CS standard limit. The isoresponse curve (Figure 4b) shows that increasing LF (arrows parallel to the x-axis) from 10 to 17.5% causes a slight decrease in CS for a given CIC content (isoresponse curves tend to be parallel to the x-axis), while increasing CIC reduces the CS from 20 to 14 MPa due to the dilution effect [44,52]. LF + CIC contents higher than 35% by mass cause a considerable strength decrease.

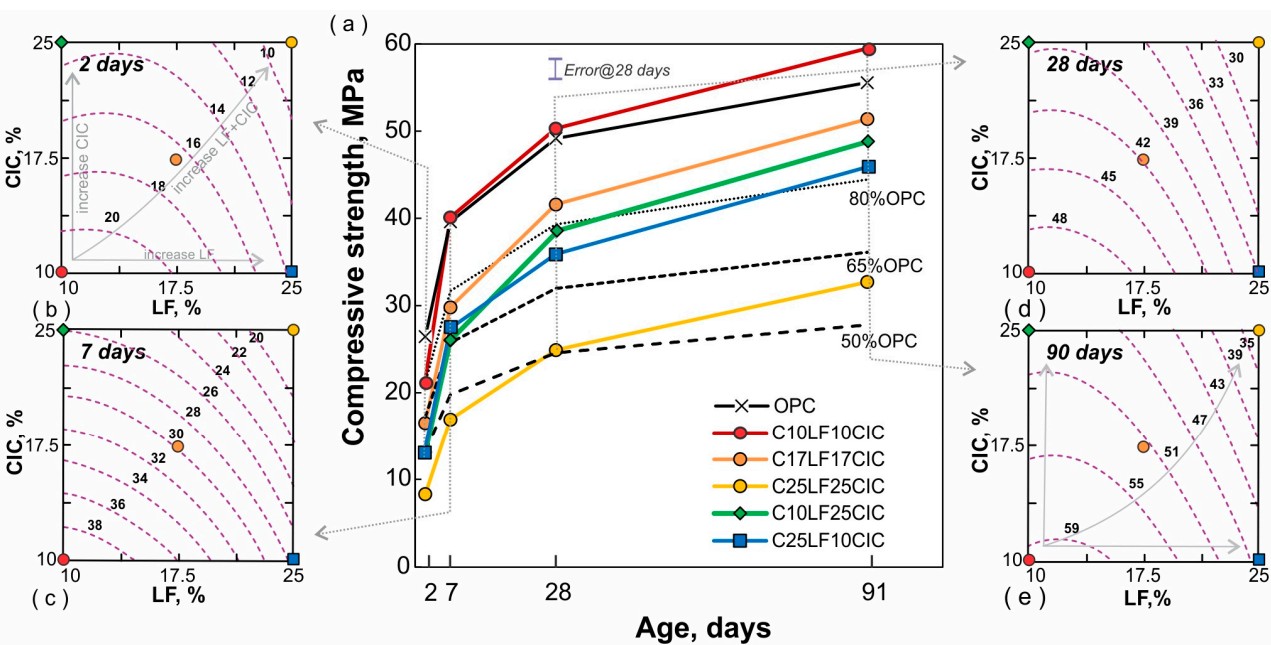

**Figure 4.** (**a**) Compressive strength of CPC and isoresponse curves at (**b**) 2 days; (**c**) 7 days; (**d**) 28 days; (**e**) 90 days.

After 7 days, the CS maintains the same evolution; it is the highest for C10LF10CIC (40.1 MPa) and equivalent to that of OPC (39.6 MPa). This is followed by mortars with 35% LF + CIC, with C17LF17CIC being the highest among them (29.8 MPa). The isoresponse curves (Figure 4c) show a quasilinear interaction between the LF and CIC, revealing that the filler effect decreases after 7 days, and the CIC does not yet contribute to the CS.

After 28 days, the CS of C10LF10CIC (50.3 MPa) exceeds that of OPC (49.2 MPa), and the standard limit for the 42.5 strength class. CPCs with 35% SCM reach 40 MPa, and CS depends on the SCM combination; the pozzolanic reaction is more evident in C10LF25CIC than in C25LF10CIC, increasing the CS. For C25LF25CIC, CS is lower than the standard limit for a 32.5 strength class (24.9 MPa). The isoresponse curves (Figure 4d,e) become parallel to the x-axis for LF < 12.5%, and the CS decreases when the LF or CIC increases. After 90 days, the three CPCs with 35% SCM achieve a compressive strength greater than 45 MPa for all combinations. Compared with OPC, CPCs with 35% SCM have a relative CS between 0.83 and 0.93.

### 3.3. Pozzolanicity

The results of the Frattini test of CPCs are shown in Figure 5. For CPCs with 10% CIC, the pozzolanicity is positive after 28 days (the points are slightly below the solubility curve), and the increase in LF increases [CaO] and decreases the [OH⁻] contents, indicating the contribution of LF to the stimulation of cement hydration. CPCs with 25% CIC are pozzolanic after 14 and 7 days to replace 10 and 25% LF, respectively. LF addition shifts the points to the left due to a reduction in the clinker factor. The early pozzolanicity of C25LF25CIC is attributed to the low clinker factor, which cannot provide enough Ca(OH)$_2$ to the system. C17LF17CIC is pozzolanic after 14 days. The CIC content provides the pozzolanicity to the CPC.

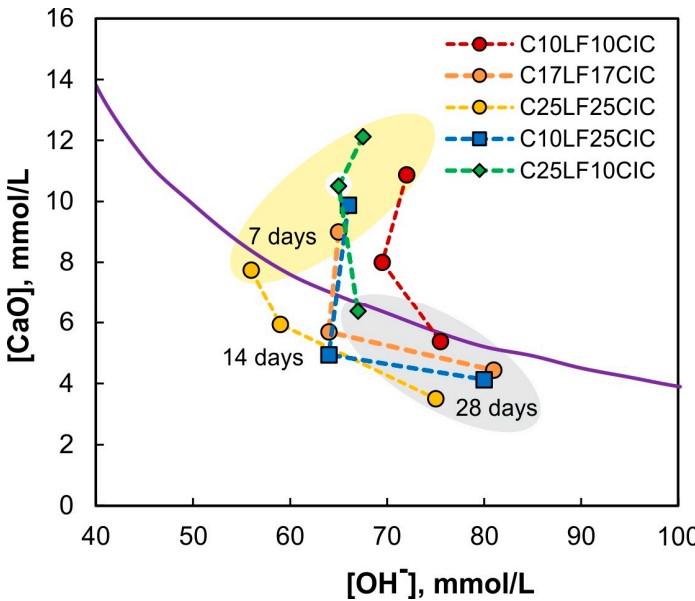

**Figure 5.** Result of Frattini's test for CPCs after 7, 14, and 28 days.

### 3.4. Hydration of Ternary Cements

The curves of the rate of heat release as a function of time at an early age are shown in Figure 6, indicating the characteristic points of (a) dissolution peak, (b) dormant period, (c) acceleration slope, (d) maximum peak, (e) sulfate depletion point, and (f) accelerated aluminate activity [53]. The CPCs show a decrease in the heat rate when the SCM replacement increases, and there are no significant changes in the time of characteristic peaks in the curve. Hence, the curve with a high heat rate corresponds to a replacement of 20% SCM (C10LF10CIC), and that with a low heat rate corresponds to 50% SCM (C25LF25CIC). CPCs with 35% SCM (C10LF25A, C17F17A, and C25F10A) show an intermediate rate of heat release. The first minimum occurs between 100 and 130 min, and the intensity decreases from 0.28 to 0.11 mW/g from low (C10LF10CIC) to high replacement (C25LF25CIC). Additionally, the dormant period is extended.

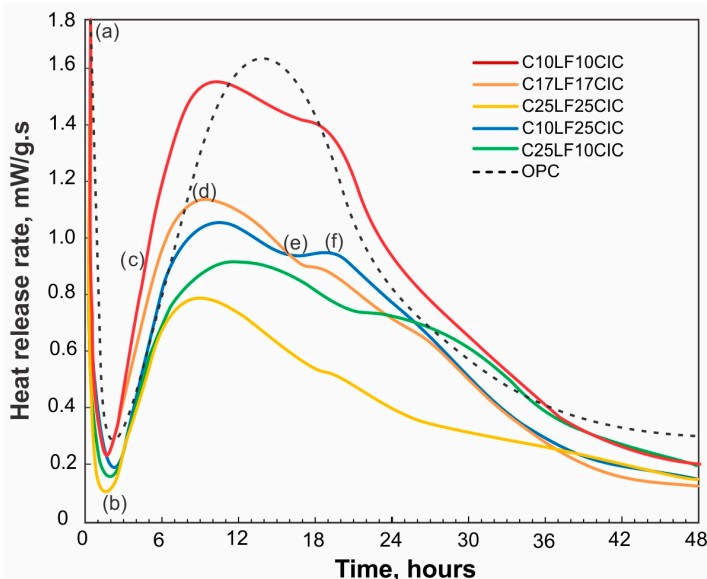

**Figure 6.** Heat of hydration for CPC and characteristic points: (a) dissolution peak, (b) dormant period, (c) acceleration slope, (d) maximum peak, (e) sulfate depletion point, and (f) accelerated aluminate activity [53].

Then, the slope of the acceleration period associated with the $C_3S$ hydration decreases with the LF + CIC content. The second peak occurs at 660 min, with an intensity of 1.57 mW/g for the C10LF10CIC, while for other CPCs, it occurs within ±60 min and with lower intensity. The reduced intensity of the peak is due to the reduction in the clinker factor. However, the acceleration rate (slope) is more significant for C10LF10CIC, followed by C17LF1CIC and C10LF25CIC, and the lowest for C25LF10CIC and C25LF25CIC, indicating that an increase in LF content from 10% to 25% for the intergrinding CPC can cause coarse clinker grains.

The sulfate depletion point occurs before the third peak representing ettringite formation. The third peak occurs at 1000, 1200, 1100, 1400, and 1160 min with intensities of 1.43, 0.94, 0.87, 0.71, and 0.50 mW/g for C10LF10A, C17F17CIC, C10LF25CIC, C25LF10CIC, and C25LF25CIC, respectively. The relative intensity of the second and third peaks increases with the CIC content (from 0.14 to 0.28 mW/g), and LF content attenuates this peak. The curves show a good relationship between the maximum peaks, indicating that the produced CPC has a good sulfate balance.

Figure 7 shows the percentage of Wn for CPCs and OPC. Dotted and dashed lines correspond to 80, 65, and 50% of Wn of the OPC. With an increase in LF + CIC from 20% to 50%, the Wn decreases due to the dilution effect. However, the observed percentage of CPCs is higher than that of clinker content (dotted and dashed lines). Wn increases up to 28 days and is then approximately constant, indicating that hydration of CIC from 28 to 90 days does not significantly increases Wn. For CPC with 35% SCM, the Wn of C25LF10CIC and C10LF25CIC shows different behavior than that of C17LF17CIC.

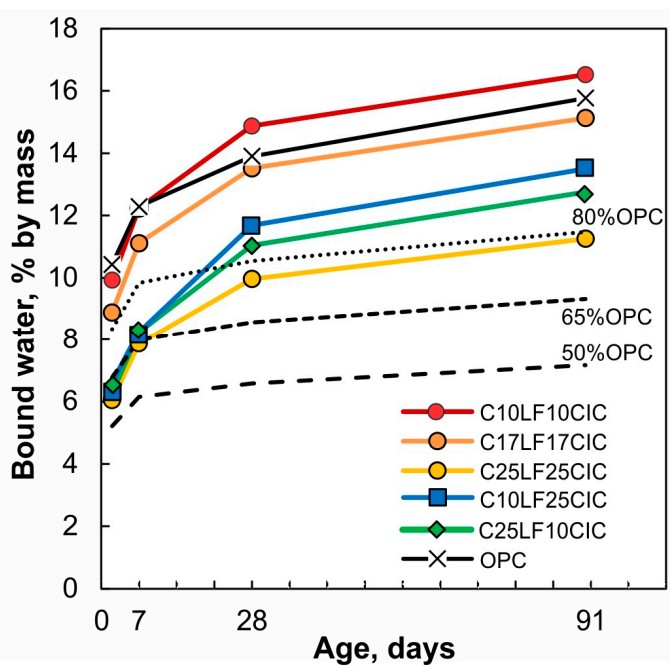

**Figure 7.** Wn in CPCs and proportional bound water of the clinker fraction (dotted and dashed lines).

*3.5. XRD*

Figure 8 illustrates the XRD patterns for CPCs after 28 days. In all pastes, ettringite, monocarboaluminate, $Ca(OH)_2$, and the peaks of unhydrated cement phases ($C_4AF$ and $C_2S$ -out of 2θ deg range in Figure 8) are clearly detected. For CPC with increasing SCM content, the intensity of ettringite decreases, and the intensity of the monocarboaluminate peak increases, while the $Ca(OH)_2$ peak present with lower intensity according to the clinker factor is reduced. However, the consumption by the pozzolanic reaction of CIC is limited at this age. For CPC with 35% SCM, the more significant change is the growing intensity of the monocarboaluminate peak when the LF increases. The filler stabilizes the ettringite and contributes to the formation of an AFm phase by the reactive alumina from

CIC, as occurred with calcined kaolinite clay [11]. Then, the hydrated compounds in CPC have the same assemblage.

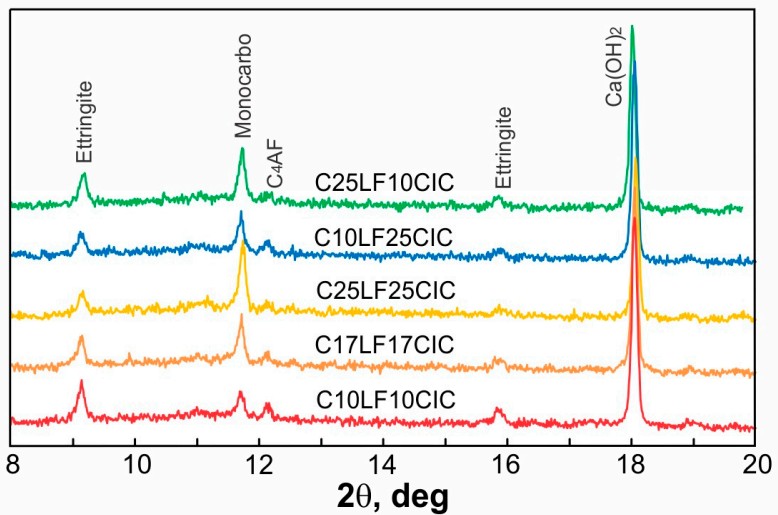

**Figure 8.** XRD patterns for CPC pastes after 28 days.

### 3.6. Performance in Concrete

All investigated concretes have good workability and cohesiveness without appreciable exudation and excellent finish. In the mix design, the concrete slump was between 15 ± 3 cm, and the flow was 45.5 ± 6.5 cm, without sign of water crown or segregation. Similar workability with the same water and admixture content appears as a target of this CPC.

Figure 9a shows the compressive strength of the concretes. As in mortar, C10LF10CIC shows the highest compressive strength (14.6 and 27.0 MPa after 2 and 7 days, respectively) but it is lower than the corresponding concrete made with OPC. For C25LF25CIC, the increase in SCM content decreases the compressive strength by 47 and 38% after 2 and 7 days, respectively. On the other hand, concretes made with CPC with 35% SCM reach a CS in a similar range: 10.6 to 12.4 and 21.4 to 22.7 MPa after 2 and 7 days, respectively.

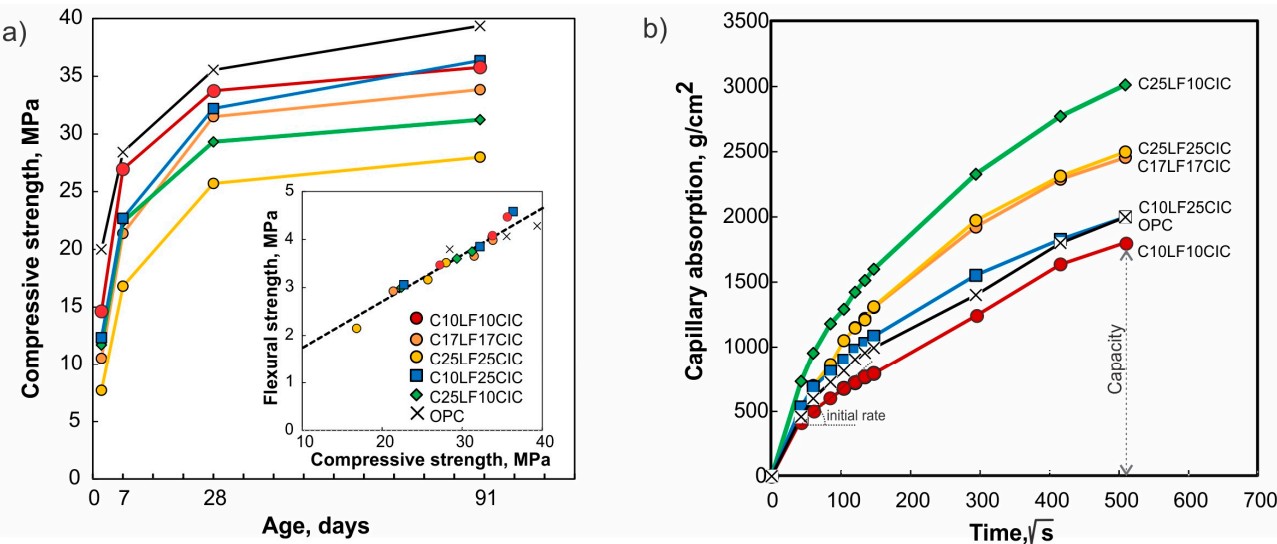

**Figure 9.** (**a**) Compressive strength of concretes: flexural vs. compressive strength. (**b**) Capillary absorption vs. the time root square for concretes after 28 days.

After 28 and 90 days, the compressive strength of C25LF25CIC is 24% lower than that of C10LF10CIC. Concretes with 35% SCM show different behavior than those registered early because strength increases as the CIC content increases. After 90 days, C10LF25CIC shows a CS of 14% (36.4 MPa) higher than that of C25LF25CIC concrete and similar to that of C10LF10CIC (35.8 MPa). At later ages, the pozzolanic reaction of calcined clay partially compensates for the dilution effect of cement caused by the non-reactive faction of CIC (~42%) and LF.

The evolution of flexural strength is similar to that of compressive strength. The relationship between both mechanical strengths for the concretes from 7 to 90 days (detail in Figure 9a) shows a good correlation.

Figure 9b shows the capillary absorption curves for concretes cured for 28 days, and Table 4 reports the initial capillary absorption rate and the capacity calculated in this test. The lowest capillary suction rate and capacity were for C10LF10CIC, while the highest values were for C25LF10CIC. At 35% SCM content, the capacity and rate of absorption increase with the LF content. For CPC with 25% CIC, the increase in LF content (10%–25%) reduces the capacity and rate of capillary absorption by 16 and 9%, respectively. Finally, C10LF10CIC and C10LF25CIC have a capillary absorption rate equivalent to that of durable concrete for rebar corrosion by carbonatation (XC4 exposure class of EN 206) elaborated with OPC.

**Table 4.** Capillary absorption rate and capacity, water penetration, and chloride migration coefficient of concretes after 28 days.

| Concrete | Capillary Absorption | | | Water Penetration under Pressure, mm | | | $D_{nssm}$ |
|---|---|---|---|---|---|---|---|
| | Initial Rate | | Capacity, | | | | |
| | $g/m^2 \, s^{1/2}$ | $R^2$ | $g/m^2$ | Average | Max | Min | $\times 10^{-12} m^2/s$ |
| OPC | 3.4 | 0.96 | 2466 | 11.0 | 13 | 8 | 12.4 |
| C10LF10CIC | 3.0 | 0.93 | 2052 | 13.6 | 18 | 10 | 15.1 |
| C17LF17CIC | 3.9 | 0.94 | 2849 | 19.8 | 24 | 12 | 20.7 |
| C25LF25CIC | 4.2 | 0.95 | 2886 | 26.2 | 40 | 17 | 29.8 |
| C10LF25CIC | 3.3 | 0.94 | 2369 | 21.4 | 30 | 15 | 12.3 |
| C25LF10CIC | 4.9 | 0.90 | 3425 | 20.4 | 25 | 13 | 17.3 |

The medium, maximum, and minimum water penetration under pressure for concretes after 28 days is reported in Table 4. The C10LF10CIC shows the lowest water penetration. In contrast, C25LF25CIC has the highest water penetration. Concretes with 35% SCM register a similar penetration (20.5 ± 0.7 mm). In all cases, the average and maximum penetration are less than 30 and 50 mm, respectively, which allows the concrete to be classified as impermeable according to the CIRSOC Code [54].

Table 4 also reports the chloride migration coefficient of CPC concretes, which are within the same order of magnitude ($10^{-12}$ m$^2$/s) and higher than that of OPC. From 20 to 50% SCMs, C25LF25CIC shows the highest $D_{nssm}$ (29.8 × $10^{-12}$ m$^2$/s), which is twice that corresponding to C10LF10CIC (15.1 × $10^{-12}$ m$^2$/s), while C17LF17CIC exhibits an intermediate value (20.7 × $10^{-12}$ m$^2$/s). For 35% SCM content, $D_{nssm}$ is reduced when the CIC content is increased, showing the ability of artificial pozzolan to reduce it [55].

## 4. Discussion

Using a RoS45 target during the intergrinding of CPCs with different proportions of LF and CIC produces a granulometric curve (Figure 2) with a predominant mode (24–27 μm), the fractional volume of which is reduced and, with fine particles increased (3–7 μm) with increasing SCM content. For multicomponent cements, grinding results depend on the grindability and volume of each component. For this system, clinker is the hardest

component, causing the clinker fraction to occupy the large particles and defining the size rejection in the separator. The higher grindability of the SCM fraction produces an increase in finer particles in the fractional volume (Figure 2) and increases the specific surface area of CPC (Table 3) cement. Increasing SCM replacement in CPC (C10LF35CIC, C25LF10CIC, and C25LF25CIC) increases the volume of finer particles and the SSA from 410 to 535 kg/m$^2$.

The packing of the CPC depends on the size and proportion of finer particles. When the volume of finer particles fills interparticle space, the packing is equal (Table 3). However, when the volume of finer particles is predominant, there is a reduction in CPC packing (Table 3), as demonstrated by Marchetti et al. [52,56]. The increase in SSA causes an increase in the water demand of CPC when packing is not reduced, defining the free water to wet the surface of particles. For CPCs, the WFT has a positive value for a w/b = 0.50 (Table 3), predicting a flowable paste [4,56]. On the other hand, SCMs have a lower density than clinker, causing a higher paste volume at constant w/b. This effect measured by the PFT (Table 3) did not change significantly for the different CPCs.

In summary, CPCs produced by intergrinding have a higher SSA, but the clinker particles are still in the coarse fraction. Changes in packing and density have a limited effect on the WFT and PFT values, causing slight changes in the flow of the standard mortar.

Upon the hydration of CPC, SCM contributions are analyzed as filler effects, including space-filling, the hydration stimulation of clinker particles, and the dilution effect computed as an increase in the effective w/c ratio [57]. On the other hand, SCM with pozzolanic activity reacts with Ca(OH)$_2$ released by clinker hydration to produce cementing compounds such as C-S-H or AFm phases that reduce the pore size and connectivity. In these CPCs, the LF, quartz, and hematite associated with CIC (Table 1) are considered non-pozzolanic materials. Furthermore, the amorphous phases exhibit pozzolanic activity, as demonstrated by Frattini's test (Figure 5). CIC exhibits a slow reaction, and its contribution is partial after 28 days, increasing at later ages [29,30,58].

At an early age, CPCs show a low heat rate with increased SCM replacement and no significant change in the occurrence time of characteristic points in the heat of the hydration curve (Figure 6), indicating that the dilution effect is predominant due to the reduction in the clinker factor. A lack of changes in the peak-time occurrence indicates that the reactive clinker particles are within the same size range. This is a shortcoming of CPC produced by intergrinding [59]. Additionally, by reducing the clinker factor, the dormant period is prolonged due to the reduction in critical pore concentration to start the acceleration period, as occurred in C25LF25CIC.

After 2 days, the stimulation effect on the clinker fraction of CPC produces a higher relative Wn compared to the percentage of OPC (Figure 7), and the hydrated compounds (Figure 8) are mainly ettringite and Ca(OH)$_2$ [44]. This effect can compensate for dilution when SCM replacement is low (10%–12% by mass). For CPC with LF + CIC > 15%, the dilution effect cannot be compensated by the filler effect, reducing the heat released (Figure 6) and the compressive strength (Figures 4 and 9a).

At later ages, the stimulation effect ceases, and the pozzolanic reaction of CIC is mainly responsible for the increase in Wn (Figure 7), causing an increase in CS with the increase in CIC content (Figure 4). On the other hand, the increase in LF reduces the CS of mortar. For standard mortar, the CS is related to the bound water (Figure 10a). The CPC strength class can be classified as 42.5 when the CIC + LF replacement is limited to 35% and 32.5 for higher replacements. Considering the compressive strength after 90 days, the progress of the pozzolanic reaction of CIC allows for an increase in the SCM replacement for a given strength class with the precaution of limiting the LF content.

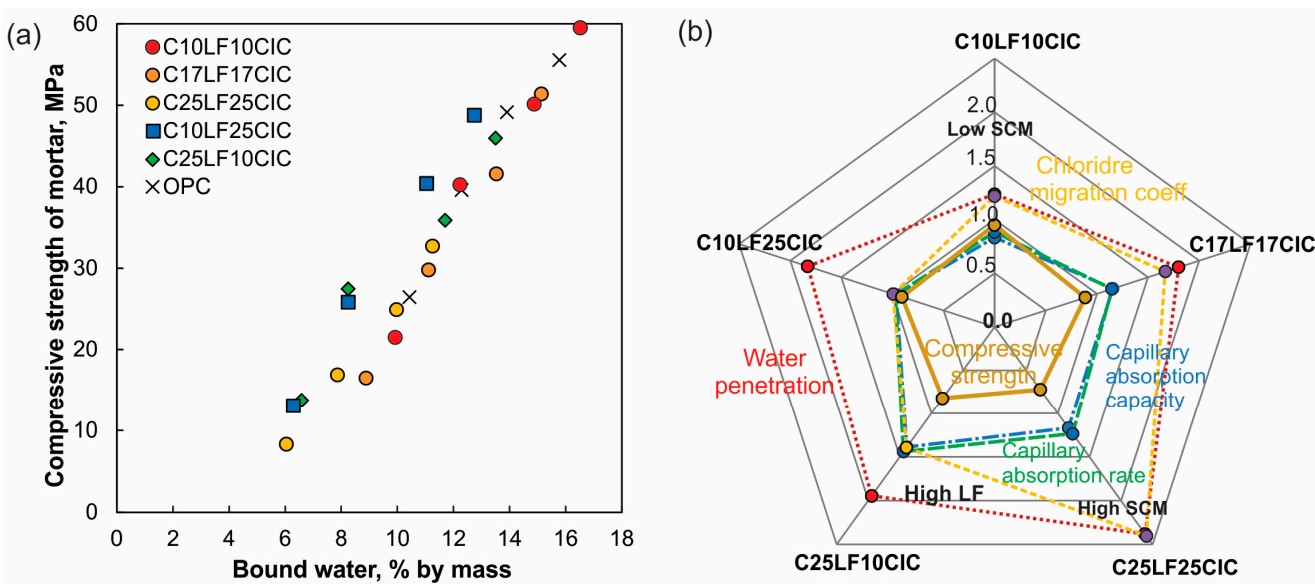

**Figure 10.** (**a**) Relationship between bound water and mortar compressive strength. (**b**) Comparison of durability parameters of concretes made with CPCs to those of OPC cements.

In concrete, the investigated CPCs have good workability as measured by slump and flow, without an extra dosage of SP as occurred in concretes made with CIC compared with calcined kaolinite clay [60]. The compressive strength of concrete (Figure 9a) shows a lower value than that of standard mortar (Figure 4), with the difference increasing when concrete compressive strength exceeds 30 MPa. This is attributed to a high proportion of micaceous minerals in the coarse granite aggregates [61], causing wicks in the concrete. The flexural strength of CPC with 25% CIC also contributes improved cement–aggregate interfaces [62].

Figure 10b shows the durable parameters of CPC concretes compared to OPC. After 28 days, all CPCs show a higher capillary absorption rate, depth of water penetration under pressure, and increased $D_{nssm}$ with increasing SCM content, all of which are more significant when the LF content increases. This behavior is attributed to the incipient pozzolanic reaction of CIC after 28 days, which is not yet sufficient to reduce total porosity, pore size, and the connectivity of the pore network, favoring water transport in the concrete matrix [63]. At later ages, the progress of the pozzolanic reaction of CIC can improve the durable performance described for binary cements [55]. The contribution of CIC to sulfate resistance [64,65] and to mitigation of the alkali–silica reaction [66] has been proven, but new studies are being conducted to determine its effectiveness in combination with LF.

## 5. Conclusions

The CPCs produced by intergrading containing clinker Portland (45%–75%), calcined illite clay (10%–25%), limestone filler (10%–25%), and gypsum (5%) exhibit a particle size distribution curve with a decrease in the size mode (25–30 μm) attributed to the particles of the clinker fraction, as well as an increase in fine particles (5–7 μm) contributed by the SCM fraction. The specific surface area increases; however, the packing and density changes do not significantly impact the WFT and PFT values. Then, the produced cements show a low water requirement measured by a standard mortar flow.

For LF + CIC replacements up to the standard limit in the composite Portland cement (35%), CPCs can be classified within the 42.5 MPa strength class. The proportions of LF and CIC are limited by compressive strength after 28 days.

The CPC hydration at an early age showed a predominant dilution effect without significant change in the times of occurrence of the characteristic points and with less intensity. After 2 days, the evolution of the Wn content increases due to hydration progress, which is a good indicator of the mortar strength development of CPC.

For intergrinding, the CPC was allowed to produce workable concretes without increasing the dose of SP, with the evolution of compressive strength equivalent to mortar up to 30 MPa, beyond which the strength is affected by the mica content in the coarse aggregate. The durable parameters of CPC concretes after 28 days show worse performance than those of OPC concrete due to the slow pozzolanic reaction of CIC, which can be improved by extending the curing time. Using new SCMs requires testing of their performance in all concrete aspects to evaluate their potential feasibility. CPCs with LF and CIC can be recommended as general-use cement in low-aggressive environments.

**Author Contributions:** Conceptualization, E.F.I. and V.L.B.; Methodology, E.F.I. and V.L.B.; validation, E.F.I. and V.L.B.; Formal analysis, E.F.I., V.L.B., G.P.C., V.F.R., C.C.C. and H.A.D.; Investigation, E.F.I., V.L.B., G.P.C., V.F.R., C.C.C. and H.A.D.; Data curation E.F.I., V.L.B., G.P.C., V.F.R., C.C.C. and H.A.D.; Writing—original draft, E.F.I., V.L.B. and G.P.C.; Writing—review and editing, E.F.I. and G.P.C.; supervision, E.F.I., V.L.B. and V.F.R.; project administration, E.F.I. and V.F.R.; funding acquisition, E.F.I. All authors have read and agreed to the published version of the manuscript.

**Funding:** This research was funded by the Agencia Nacional de Promoción de la Investigación, el Desarrollo Tecnológico y la Innovación (grant number PICT 2018-3405).

**Institutional Review Board Statement:** Not applicable.

**Informed Consent Statement:** Not applicable.

**Data Availability Statement:** No data available.

**Acknowledgments:** The authors also are grateful to Loma Negra CIASA for the materials and data supplied during the industrial pilot trial.

**Conflicts of Interest:** The authors declare no conflict of interest.

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
