# Peer review of "Performance of Composite Portland Cements with Calcined Illite Clay and Limestone Filler Produced by Industrial Intergrinding"

_minerals, doi:10.3390/min13020240_

Round 1

Reviewer 1 Report

This manuscript reports of research on calcined illite clay as an additive to cement. This is a topic of both technological and ecological relevance. The review of the related literature is well developed and provides a good introduction to the experimental part. The work plan is also correct and well presented. Notable is the clear graphic design of the work. The description of experimental methods is mostly accurate. I only suggest authors to complete the error bars in Fig. 4 and carefully check the paper again for typos. It would also be useful to supplement the work with full XRD analysis - perhaps as part of supplementary materials. An interesting continuation of the research (not necessarily within this manuscript) would be to supplement it with a complete set of structural analyses, such as FT-IR and SEM. These observations allow the reviewer to recommend the paper for publication after minor revisions.

Author Response

Thank you very much for the review of the manuscript and your comments about our paper.  In figure 1, we have incorporated the error bar for the compressive at 28 days which is ±1.8 MPa as requested.  Concerning future works, they are underway, and we hope to set them in this or another journal in a short time.

Reviewer 2 Report

This study provides useful information for the manufacture of sustainable cement or concrete. However, since the manuscript contains some errors, it should be corrected carefully. 

(Title) Is composite Portland cements (CPC) a commonly used term? I think blended cement is a more appropriate term. Check it. 

(line 36) What is the full name of GCCA? Is there any reason why it should be abbreviated? 

(line 95) The illite content in the raw clay is key information, but is not presented. 

(Table 1) The position of “CIC” and “limestone” should be reversed. Also, can the authors be sure that all amorphous phases in CIC are all meta-illite? i.e., is there any possibility that the raw material contains kaolinite? 

(line 105) On what basis was the calcination temperature (950 ) determined? 

(line 194) How was the weight loss due to decarbonation of the raw material (i.e., limestone) excluded? 

(line 241) Express the specific surface area in formal units (cm2/g or m2/g). 

(Figure 5) The title of X-axis needs to be corrected (OH-). 

(Figure 7) This result is inaccurate unless the CO2 content in limestone has been precisely excluded. 

(Figure 9a) The sample name of the blue square marker should be changed to C10LF25. 

(line 437) As mentioned earlier, specify the unit of SSA accurately. 

(5. Conclusion section) All the results and conclusions derived are inevitably different depending on the illite content in the raw (uncalcined) clay. In this respect, the illite content in the clay used should be provided.

Author Response

Comments and Suggestions for Authors

This study provides useful information for the manufacture of sustainable cement or concrete.  However, since the manuscript contains some errors, it should be corrected carefully.

Thank you very much for revising our paper and the corrections that help improve our manuscript so that its reading is more straightforward.  Your remarks or suggestions have been included in the manuscript as described here:

(Title) Is composite Portland cements (CPC) a commonly used term?  I think blended cement is a more appropriate term.  Check it.

According to IRAM 50000 (in the same way as EN 197-1:2011 Cement - Part 1: Composition, specifications and conformity criteria for common cements), composite portland cements (CPC) are defined as a mix of two or more SCMs.  There are two types of cements with two SCMs ranging from 6 to 20 % by mass (CEM II A-M ) or 21 to 35 % by mass (CEM II A.-B), including four blended cements explored in this paper.  The extra point is of experimental design a composite cement (C25LFC25CIC)  is covered by the EN 197-5:2021 Cement - Part 5: Portland-composite cement CEM II/C-M and Composite cement CEM VI.

Then, the CPC used is a more accurate definition than blended cement.

(line 36) What is the full name of GCCA?  Is there any reason why it should be abbreviated?

The full name of GCCA was included.

(line 95) The illite content in the raw clay is key information but is not presented.

The XRD and mineralogical composition of illite was included in this phrase “Figure 1 shows the XRD pattern of raw illite claystone and the average mineralogical composition determined during the industrial trial pilot using the Rietveld method was illite (60%), quartz (34%); feldspars (3%); hematite (1.5%) and less than 1% of chlorite; kaolinite and/or calcite.

(Table 1) The position of “CIC” and “limestone” should be reversed.  Also, can the authors be sure that all amorphous phases in CIC are all meta-illite?  i.e., is there any possibility that the raw material contains kaolinite?

The Limestone and CIC titles in Table 1 were changed.  The XRD pattern of raw claystone reveals only traces of chlorite and kaolinite.  Then, we think the CIC composition is an amorphous phase mainly composed of meta-illite at this calcination temperature.  When the temperature increases, the glass formation is promoted (Irassar et al.; Calcined Illite-Chlorite Shale as Supplementary Cementing Material: Thermal Treatment, Grinding, Color and Pozzolanic Activity.  Appl. Clay Sci. 2019, 179, doi:10.1016/j.clay.2019.105143)., but we can not separate the meta-clay and glass formation.  Then, we called this phase amorphous.

(lin 105) On what basis was the calcination temperature (950 ℃) determined?

Laboratory studies previously reported are referenced.  “The calcination temperature was determined in laboratory studies previously reported for different compositions of illite claystone [29, 30].” Several variables, such as temperature, residence time, rotation speed, flame configuration, and gas-oxygen mixture, were studied during the industrial pilot trial.  This temperature was selected for producing the calcined clay for use as SCM in cement based on thermal transformation and production considerations.

(line 194) How was the weight loss due to decarbonation of the raw material (i.e., limestone) excluded?

For each CPCs, the LOI was determined for unhydrated cement, and it was discounted as described in the procedure of Wn reported by Powers (Powers, T.. . The Non-Evaporable Water Content of Hardened Portland Cement Paste. ASTM Bull. 1949, 68–75).

(line 241) Express the specific surface area in formal units (cm2/g or m2/g).

In the metric system, the specific surface area is expressed as m2/kg as indicated by the cement standard.

(Figure 5) The title of the X-axis needs to be corrected (OH-).

The title was corrected, and the figure was changed

(Figure 7) This result is inaccurate unless the CO2 content in limestone has been precisely excluded.

We agree that DTA-TG obtains more accurate results.  Still, this method indicates the progress of hydration in all types of cement.  The mass loss due to the decarbonation of limestone was excluded as the LOI in the CPC.

(Figure 9a) The sample name of the blue square marker should be changed to C10LF25.

The figure was revised and changed

(line 437) As mentioned earlier, specify the unit of SSA accurately.

In the metric system, the specific surface area is expressed as m2/kg as indicated by the cement standard.

(5.  Conclusion section) All the results and conclusions derived are inevitably different depending on the illite content in the raw (uncalcined) clay.  In this respect, the illite content in the clay used should be provided.

Conclusions are based on the performance of this experimental program with calcined clay obtained from raw claystone containing more than 50% of illite.  Also, we studied the limitation of raw materials with a low proportion of illite (<40%) that give poor pozzolanicity due to the large proportion of quartz, but it is out of this paper.

Round 2

Reviewer 2 Report

This manuscript is well prepared to be accepted for publication.